# Training dynamics impact post-training quantization robustness

**Albert Catalan-Tatjer**[†‡*]  **Niccolò Ajroldi**[†§]  **Jonas Geiping**[†‡]

[†]ELLIS Institute Tübingen
[‡]Max Planck Institute for Intelligent Systems & Tübingen AI Center
[§]OpenEuroLLM

## Abstract

While post-training quantization is widely adopted for efficient deployment of large language models, the mechanisms underlying quantization robustness remain unclear. We conduct a comprehensive analysis of quantization degradation across open-source language model training trajectories up to 32B parameters and 15T training tokens to accurately assess the relationship between training dynamics and quantization performance. Our key finding is that quantization errors in large-scale training runs are driven by a complex interplay between learning rate and other training hyperparameters. Specifically, once learning rates decay, validation loss and quantization error diverge, largely independent of training data scale. To investigate interventions on the training dynamics and identify specific configurations that can modulate quantization robustness favorably, we train our own models in controlled experiments up to 100B tokens, and analyze how the loss curvature evolves and interacts with the learning rate during training. Our results challenge the assumption that increasing dataset scale inherently compromises quantization effectiveness, demonstrating instead that strategic training hyperparameter interventions can improve quantization quality at scale.

## 1 Introduction

Deep learning has already entered the low-bit era (NVIDIA, 2025). This transition has been enabled by specialized hardware support and algorithmic innovations, with quantization serving as the core technology driving low-precision workloads. Modern neural networks are surprisingly *quantizable*, and even modern large language models (LLMs) trained over trillions of tokens in 16 and 32 bits of precision can be quantized into a zoo of low-bit formats, leading to a widespread adoption throughout the entire model deployment workflow, and large interest from both hobbyists and model service providers. In the following we will denote this workflow as *post-training quantization* (PTQ).

Generally, quantization maps models trained with high-precision formats to lower-precision representations. Common strategies to preserve performance involve scaling (Xiao et al., 2024), rotating (Ashkboos et al., 2024), grouping (Lin et al., 2024), or indexing in codebooks (Tseng et al., 2024). GPTQ and AWQ (Frantar et al., 2023; Lin et al., 2024; Tseng et al., 2024) unlock low-bit primitive throughput and memory gains during inference not only through strong quantization strategies, but also through specialized kernels that support fast inference on quantized models. However, despite the widespread use of PTQ in all layers of the community, from model providers to practitioners, there is still a limited understanding of the principles that govern the brittleness of quantization, i.e. the *ease* with which different models can be quantized and what error rates to expect. Recent efforts to study quantization in Kumar et al. (2024) and Ouyang et al. (2024) suggest that PTQ becomes less effective for LLMs as training progresses, arguing that the number of training tokens relative to model size is a central factor in quantization sensitivity. Consequently, as datasets inevitably grow larger (Brown et al., 2020), they expect degradation to become more severe, ultimately questioning whether post-training quantization remains viable for future models. However, we find these results overlook a key piece of the puzzle: the influence of training dynamics on the ease of quantization.

---

[*]albert.catalan-tatjer@tue.ellis.eu

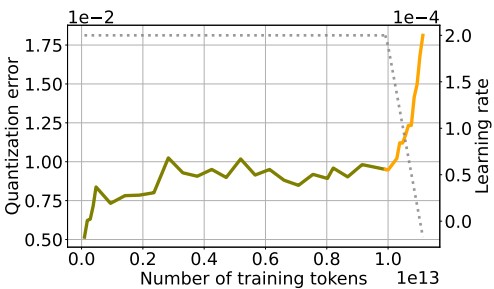 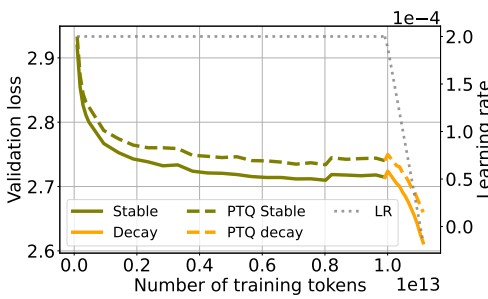

**(a)** 4-bit quantization error vs training tokens.          **(b)** Validation loss vs training tokens.

**Figure 1: Evolution of quantization error and validation loss during training of SmolLM3** (Bakouch et al., 2025). We report quantization error and validation loss throughout training under both the **constant** ($\eta = 2e^{-4}$, up to 10T tokens) and **annealing** phases of the learning rate schedule (whose evolution is shown as dotted lines). As the learning rate decays, validation loss consistently decreases, whereas quantization error rises sharply and to a much greater extent than at any earlier point in training.

While Ahmadian et al. (2023) showed that large activation outliers can be controlled with weight decay to improve PTQ performance, the effect of training hyperparameters on quantization quality has been difficult to study, since open-weights releases typically provided only a single checkpoint (Touvron et al., 2023), offering no insight into training details or into the *trajectory* of quantization error during training. However, with the recent surge of open-source large language models (LLMs) (Biderman et al., 2023; Groeneveld et al., 2024; OLMo et al., 2025; Bakouch et al., 2025), which vary substantially in training design and learning rate configurations, we now have access to much richer data to study this question in detail. Open-source model training runs document a number of hyperparameter choices, but how these choices affect quantization is rarely discussed.

In this work we provide a systematic study of the post-training quantization error across training stages for six modern, open-source LLM training efforts. While previous work has studied quantization degradation in controlled settings or for short training runs below 300B tokens, we include trajectories of open-source LLMs of up to 32 billion parameters trained on up to 15 trillion tokens. Through this investigation, we find that the actual hyperparameter choices taken by model trainers play a larger role in quantization error than previously expected. Training our own models, we verify the effect of learning rate scheduling and weight averaging on PTQ error in controlled studies, and provide actionable suggestions to intervene on quantization. In summary,

- We measure quantization error across hundreds of intermediate training checkpoints from major open-source LLM families and correlate quantization error trajectories with training stages and learning rate schedules in Section 3.
- In controlled experiments in Section 4, we verify that quantization error is modulated by learning rate schedule. Maintaining larger learning rates, all else being equal, reduces quantization error.
- Informed by these findings, we show in Section 5, that, for our own training runs, lower quantization error can be achieved by optimized learning rate schedules, and how weight averaging along training trajectories can be used to improve quantization performance.
- Finally, in Section 6, we analyze the geometric properties of the loss suggesting that the proposed interventions interact with quantization performance via the promotion of flatter minima.

Through a systematic investigation and concrete examples, we highlight that training hyperparameters, and the resulting training dynamics significantly change how easy it is to quantize modern LLMs. We argue that studying PTQ continuously during pretraining, and especially during hyperparameter selections before large-scale runs, should be an essential step, as we identify several cases, in which, for example two learning rate choices seemed equally promising, but choosing the smaller one, did lead to an increased quantization error down the line.

## 2 BACKGROUND AND RELATED WORK

### 2.1 POST-TRAINING QUANTIZATION

Post-training quantization methods reduce the memory required to run large neural networks by reducing their numerical precision. However, as LLM inference is dominated by auto-regressive decoding, which is in turn limited by memory bandwidth (the rate at which model weights can be

transferred to an accelerator's compute units, e.g. streaming multiprocessors on GPUs), quantization can often improves the speed of the model.

The most naive quantization method is to simply cast all floating-point parameters of the model to the desired precision. More advanced algorithms, such as BNB, AWQ, or GPTQ (Frantar et al., 2023), optimize which parts of the model to quantize and by what approach to minimize errors, when quantizing **weights**, **activations** and **KV-cache**. In particular, for a linear layer with weights $W$, let $X$ denote the input and $W_Q$ the quantized low-precision weights derived from $W$ by some method. During inference, $W_Q$ is loaded onto the GPU and the matrix multiplication (GEMMs) is performed with the dequantized weights $\hat{W}$ such as $X\hat{W}^T$. For weight and activation quantization, the input $X$ is also quantized. Modern mixed-precision kernels fuse the dequantization and multiplication steps for efficiency. Initially, quantization methods would aim to minimize the **weight error** $||W - \hat{W}||$ (Courbariaux et al., 2016); however, more recent approaches minimize the **reconstruction error** $||XW^T - X\hat{W}^T||$. The latter methods require a calibration dataset to compute $X$ at quantization time, several other variants exist (Frantar et al., 2023; Lin et al., 2024; Tseng et al., 2025)

Most quantization approaches build upon variations of these core concepts (Vanhoucke & Senior; Jacob et al., 2017; Tseng et al., 2024; Dettmers et al., 2022; Ashkboos et al., 2024): high-precision auxiliary states, such as scaling factors, to map between the dynamic range of original tensors and that representable in low-precision; dividing the quantization problem into smaller groups of typically 128 weights; processing outliers that would affect the dynamic range of the group with different strategies. While numerous quantization techniques exist in the literature, we focus our analysis on GPTQ (Frantar et al., 2023) quantization at 3- and 4-bit precision levels. However, our supplementary experiments demonstrate that AWQ (Lin et al., 2024) and BitsAndBytes (BNB) Dettmers et al. (2022) quantization methods exhibit analogous trends, as detailed in Appendix A.

## 2.2  LLM Training Hyperparameters

Large-scale pretraining of neural networks, such as language models, is dependent on a large number of hyperparameter choices. We review here some fundamental elements of the pretraining pipeline, as we later show they are linked to quantization error and can be exploited to modulate it.

A key aspect of optimization is the choice of a **learning rate schedule**. Whereas earlier language model training largely relied on **cosine decay** schedules (Loshchilov & Hutter, 2017), more recently model builders have shown increasing interest in the trapezoidal schedule (Zhai et al., 2022; Hu et al., 2024), also known as Warmup–Stable–Decay (**WSD**). This scheme splits training into a constant learning rate phase followed by a linear-decay stage, enabling training across different compute budgets with significantly fewer resources (Haegele et al., 2024) and has hence seen growing adoption (Bakouch et al., 2025; Nezhurina et al., 2025; Apertus Team, 2025). Alongside the scheduler shape, the **peak learning rate** (LR) itself is arguably one of the most important parameters for final model performance (Tissue et al., 2024) and training stability (Wortsman et al., 2023). Together with the peak LR value, the value after annealing can also impact performance (Bergsma et al., 2025), scaling law derivation (Li et al., 2025) and adaptability to supervised finetuning (Singh et al., 2025). Overall, many design choices remain somewhat arbitrary, frequently guided by heuristics (OLMo et al., 2025) and often yielding equivalent results when sufficiently tuned (Haegele et al., 2024). In this work, we argue that one additional line of analysis should be **robustness to quantization**, as the interplay between these variables and PTQ degradation reveals underexplored design decisions and a path for guiding future choices.

## 2.3  Model brittleness to post-training quantization

How well will a certain quantization algorithm work for a given, already trained, LLM, and does this depend on the size of the model, or the amount of training data? Recently Kumar et al. (2024) and Ouyang et al. (2024) developed scaling laws for quantization error, in which they relate the scale of training dataset with the degradation induced by quantization. In summary, they reach a similar conclusion, **as models are trained on more data, they exhibit higher quantization induced degradation.** However, scaling up the training dataset is one of the primary levers to improve model performance, and small overtrained models are becoming increasingly popular (Gadre et al., 2024).

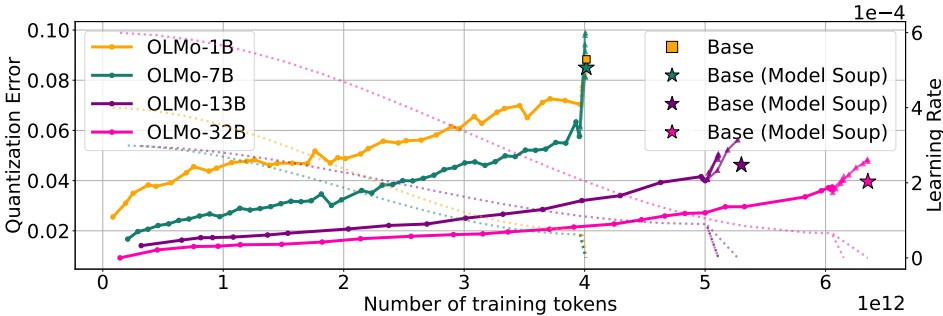

**Figure 2: 3-bit quantization error along the training trajectories of OLMo2 models**. Error grows gradually during cosine decay but spikes under the steep linear decay phase. Model souping (⋆) reduces degradation, achieving lower PTQ error than any individual run.

Yet, these studies overlook the role of the training dynamics in model robustness to post-training quantization. In fact, we find that on open sourced LLMs, quantization degradation abruptly increases as learning rates decays, regardless of training data size. In Section 4 we investigate these contradicting results and we find that their characterization of the effect of training dataset scale and quantization performance is mostly confounded by the learning rate hyperparameters used in their experiments. Overall, we identify this gap in the literature and address this crucial question: *what is the relationship between the training dynamics and quantization performance?*

## 3 Post-training quantization of models in the wild

In this section, we analyze training trajectories of the following models: OLMo model family (1B, 7B parameters; trained on 2.5T-3T tokens) (Groeneveld et al., 2024); OLMo2 family suite (1B, 7B, 13B, 32B; 4TT–6TT) (OLMo et al., 2025); SmolLM3 (3B, 11TT) (Bakouch et al., 2025); Apertus (8B, 15TT) (Apertus Team, 2025); Open-science (1.3B, 1TT) (Nezhurina et al., 2025), for which we consider the Nemotron-cc release (Su et al., 2025); and Amber (7B, 1.3TT) (Liu et al., 2023). We use GPTQ (Frantar et al., 2023) to post-train quantize them to 3 and 4 bits. We detail the quantization process in Appendix A, and share the complete set of results for all model families in Appendix B.

We evaluate PTQ robustness by first examining quantization error in validation loss and later by assessing its impact on downstream tasks.

### 3.1 Quantization-Induced Degradation on Validation Loss

To more accurately represent the intuition that increases in cross-entropy loss are more expensive the lower the cross-entropy is (as loss decrease is roughly logarithmic in compute), we show relative cross-entropy loss, defined as $\left(\frac{\text{CE}(\hat{W})}{\text{CE}(W)}\right) - 1$.

We decouple the effect of learning rate decay from the amount of training data consumed, we first focus on models trained with a **Warm up–Stable–Decay** schedule. We begin by examining Figure 1a, which shows quantization error alongside the learning rate during the training trajectory of **SmolLM3**. We observe that, while quantization error increases rapidly in the beginning of training, it stays relatively constant during the 11 trillion tokens of stable phase, and only as the learning rate decays does quantization error spike. Figure 1b shows how the validation loss follows a similar—albeit inverse—curve than that of the quantization error. Similarly, **OpenSci** training runs from Nezhurina et al. (2025) in Figure 11 display an analogous pattern: quantization error surges sharply as the learning rate decreases, for the different models on vastly different token budgets.

Next, we consider the **OLMo2** model family, which includes four language models with 1, 7, 13, and 32 billion parameters, all developed using a consistent training methodology. Training occurs in two phases: an initial general pretraining phase using 4-6 trillion tokens with **cosine** learning rate decay, followed by a second phase that applies a short and sheer linear decay schedule across different orders of high-quality data configurations, also referred to as "ingredients". The final

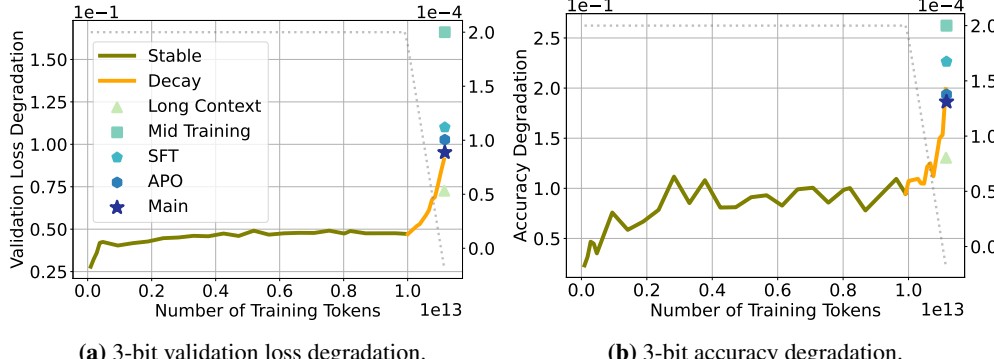

**(a)** 3-bit validation loss degradation.     **(b)** 3-bit accuracy degradation.

**Figure 3: 3-bit quantization effects across SmolLM3 post-training stages**. Degradation in validation loss (left) and downstream accuracy (right) show that PTQ effects differ across stages and appear sensitive to post-training interventions. The final model, a weighted average of mid-training and APO, shows better robustness than both individual components.

model weights are obtained through model souping (Wortsman et al., 2022), averaging models trained with different ingredients, except for the 1B parameter model, which retains weights from a single decay trajectory. Figure 2 presents quantization error and learning rate trajectories for the four models. The quantization error shows a different trend across the two phases, increasing gradually during slow cosine decay, but rising sharply under steep linear annealing. Although the learning rate itself *may not directly cause this degradation*, this observation once again suggests a deeper connection between optimization dynamics and quantization performance. Finally, we report the quantization error for the model soup, and find that averaging substantially reduces degradation, with the model soup achieving lower PTQ error than any of the individual ingredients. We will return to this observation later in Section 4 and 5.

## 3.2 QUANTIZATION-INDUCED DEGRADATION ON DOWNSTREAM TASKS

While cross-entropy loss serves as a convenient proxy for model performance, downstream evaluation better reflects the practical utility of a model. Following OLMo et al. (2025), we evaluate performance across 12 established benchmarks and report the average 5-shot accuracy across all tasks (see Appendix D for additional details on the evaluation pipeline).

In Figure 3 we show the performance degradation induced by 3-bit quantization on SmolLM3. Alongside the validation loss (Figure 3a), we present the relative accuracy drop, defined as $\frac{Acc(W)-Acc(\hat{W})}{1-Acc(W)}$ (Figure 3b). Despite fluctuations, a similar pattern can be identified in both curves: performance degradation increases as the learning rate decays. We observe similar results across individual tasks and report them in Appendix D (Figure 17, Figure 18).

Modern LLMs are optimized beyond general pretraining stages to promote alignment, extend context, incorporate supervised fine-tuning, and perform instruction tuning (Tie et al., 2025). Here, we study the effect of quantization across **post-pretraining** stages. In SmolLM3, these include *long context* training, a *mid-training* phase to incorporate general reasoning capabilities, *supervised fine-tuning (SFT)* for domain-specific skills, and *anchored preference optimization (APO)* (D'Oosterlinck et al., 2024) to promote alignment. Finally, the released *(main)* model is a linear merge with weights of 0.9 and 0.1 of the APO model and a mid-training checkpoint. Figure 3 reports the performance degradation under 3-bit quantization after each stage in SmolLM3. Interestingly, context extension sensibly reduces quantization degradation, while mid-training largely amplifies it. PTQ degradation then decreases through SFT and APO. Remarkably, although the main model is obtained by averaging the mid-training and APO weights, it exhibits lower quantization degradation than either of them individually. We recall similar results from the previous analysis on OLMo-2 (Figure 2), where model soups across data mixtures exhibited lower quantization degradation than any of the individual components. These results suggest that averaging benefits quantization, a novel finding we investigate further in Section 5.

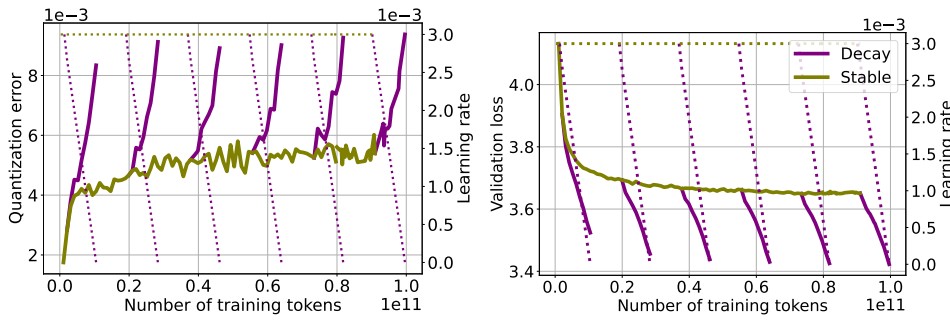

**(a)** 4-bit quantization error vs training tokens.  **(b)** Validation loss vs training tokens.

**Figure 4: 4-bit quantization error at different training durations**. We use WSD, training a 160M-parameter transformer up to 100B tokens and performing additional cooldowns at 12B, 28B, 46B, 64B, 82B tokens. Figure 4a shows quantization error during training with different token budgets, and Figure 5b the corresponding validation loss. Despite varying the amount of training data, all runs show comparable quantization error after cooldown, highlighting that error spikes are associated with training dynamics rather than token budget.

## 4 CONTROLLED EXPERIMENTS

### 4.1 REPLICATING THE OBSERVED PHENOMENA

To understand the insights from Section 3, we conduct pretraining experiments with transformer models on a smaller scale, varying token budget, learning rate, LR schedule, and weight decay one at a time. We follow Biderman et al. (2023) for model specifications, and FineWebEdu (Penedo et al., 2024) as pretraining corpus (see Appendix C for details on the training procedure and hyperparameters). We use GPTQ, and discuss results for additional quantization methods in Appendix A.

In Figure 4 we show quantization error and validation loss across a range of token budgets, which we obtain by decaying the learning rate at different steps during training. We observe that the constant learning rate stage is not immune to PTQ degradation, showing a slight increase in quantization error. At the same time, despite training durations ranging from 10B to 100B tokens, models achieve *comparable quantization error* after decay, which spikes as learning rate decays and validation loss drops, regardless of than token count. In Figure 21 we replicate the experiment using a cosine decay schedule, where model performance (Figure 21b) and quantization robustness (Figure 21a) vary with the training horizon. However, changes in the peak learning rate, and thus the scheduler shape, have a larger impact, in some cases yielding improved quantization error at lower validation loss.

In conclusion, this evidence suggests that the phenomena observed in Section 3 are not merely serendipitous outcomes of complex model interactions, but are strongly shaped by training dynamics, with factors such as learning rate decay playing a key role in quantization performance.

### 4.2 SCALING TRENDS IN PRIOR WORK ARE DOMINATED BY LEARNING RATE SCHEDULES

In an effort to explain the rise of quantization error during training, previous studies attributed this phenomenon to dataset size or training duration, concluding that *PTQ degradation increases as models are trained on more data* (Kumar et al., 2024), and hence that quantized undertrained models scale more favorably (Ouyang et al., 2024). We argue that these works did not sufficiently control for a key confounder, namely the optimization dynamics induced by the learning rate schedule, which we find to be the primary driver of their observed degradation.

Specifically, we replicate analyses from Kumar et al. (2024) in Figure 5, training models at different token budgets under both original cosine schedule and WSD schedule. While cosine results (blue) suggest that $\delta_{PTQ}$ increases noticeably with token budget, we show that a comparable WSD schedule (brown) can yield lower validation loss, with degradation growing more slowly (70M) or remaining stable (160M), indicating that the effect cannot be ascribed to data alone (see also Figure 21 for a similar conclusion).

Finally, we argue for additional caution when collecting checkpoints at different token counts, as done in Ouyang et al. (2024). We recall that similar considerations have been discussed in the scaling law literature: Hoffmann et al. (2022) suggested that their power law discrepancy with Kaplan et al.

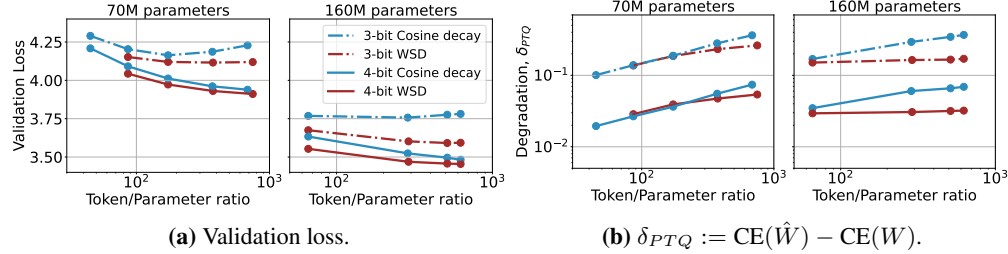

**(a)** Validation loss.

**(b)** $\delta_{PTQ} := \mathrm{CE}(\hat{W}) - \mathrm{CE}(W)$.

**Figure 5: Learning rate affects quantization scaling trends**. Following Kumar et al. (2024), we train 70M and 160M transformer models with cosine decay across different token budgets, and a WSD schedule under the same model configurations. Cosine decay replicates prior results, with $\delta_{PTQ}$ increasing at larger token budgets, while WSD shows slower growth at 70M and no increase at 160M, hinting that other factors beyond data volume influence quantization scaling.

(2020) arose from differences in learning rate schedules, and further works validate the importance of collecting checkpoints only after learning rate annealing (Haegele et al., 2024). We suggest that the same discretion is necessary when deriving scaling laws for quantized models, as optimization dynamics influence observed robustness (Figure 4).

## 5 INTERVENTIONS ON THE TRAINING DYNAMICS

Having explored the connection between training dynamics and quantization degradation we investigate how simple interventions can modulate PTQ robustness and achieve better quantized models.

### 5.1 LEARNING RATE

In Figure 6, we demonstrate how different peak learning rates impact quantization. Figure 6a shows that higher learning rates consistently lead to smaller errors, with curves inversely ordered by rate magnitude. Figures 6b and 6c report full-precision versus 4-bit and 3-bit quantized validation losses. These parametric curves capture quantization error relative to total validation loss: perfect quantization would lie on the $x = y$ bisector, with deviations measuring the error. Comparing curves with LR $1\mathrm{e}{-}3$ and $3\mathrm{e}{-}3$ shows that, at similar validation loss, the larger rate achieves better low-bit quantization, at no apparent cost. This suggests that, for comparable full-precision performance, employing a larger learning rate might be preferable, as it enhances low-bit quantization performance. We replicate this experiment on a 300B token pretraining run of OLMo2-7B in Figure 23.

Learning rate schedules designate the magnitude of the learning rate throughout training, represented as dotted lines in Figure 22a. On one hand, while the cosine schedule (green) has a much higher peak learning rate, its profile is dominated by the one of WSD decay phase (yellow and blue). Despite this rapid decay, the cosine schedule still achieves lower quantization error and better validation loss than the WSD schedule. This indicates that quantization performance depends on training dynamics beyond just the learning rate magnitude at any single point. On the other hand, examining 3-bit quantization in Figure 22c reveals that cosine schedules experience sharp upward curvature near the end of training, likely due to very small learning rates in the final steps. This suggests that cosine schedules' inability to control end-of-training learning rates, where the rate becomes small regardless of the initial peak, may hurt quantization performance compared to schedules like WSD that maintain better control throughout training.

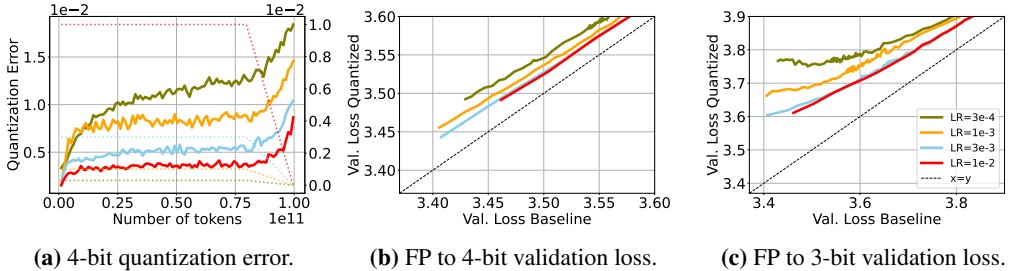

**(a)** 4-bit quantization error.

**(b)** FP to 4-bit validation loss.

**(c)** FP to 3-bit validation loss.

**Figure 6: Larger learning rates lead to lower quantization error**. Figure 6a displays the quantization error achieved by fixing the training recipe and varying the learning rate. We observe that quantization error decreases when employing higher learning rates. Furthermore, Figure 6b and 6c show that, at similar validation loss, larger learning rates achieve better low-bit quantization, at no apparent cost.

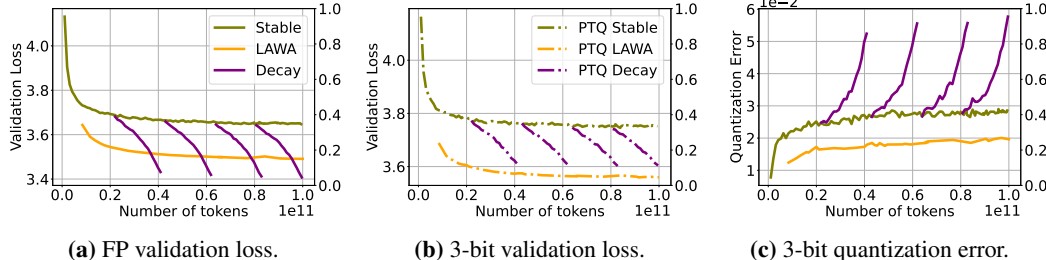

(a) FP validation loss.        (b) 3-bit validation loss.        (c) 3-bit quantization error.

**Figure 7: Weight averaging as an alternative to LR decay for PTQ**. Validation performance and quantization error for a 160M model trained on 100B tokens at constant learning rate. We compare intermediate learning rate cooldowns with weight averaging of checkpoints collected from the stable phase. We report the validation performance of the full-precision model (Figure 7a), the 3-bit quantized model (Figure 7b), and their difference (Figure 7c). Whereas LAWA falls short of learning-rate decay in the full-precision setting, its 3-bit PTQ performance yields lower validation loss than all cooldowns, demonstrating a successful setting for LAWA.

## 5.2 WEIGHT AVERAGING

Given the encouraging results on quantizing model soups in Section 3.1, and the detrimental effect of learning rate decay on quantization performance, a natural question is whether weight averaging could serve as an alternative and mitigate its negative impact[1]. Intuitively, averaging parameters along the training trajectory reduces noise and can approximate the effect of learning rate decay. Prior work derived equivalent averaging schemes for common LR schedules under SGD (Sandler et al., 2023), and later studies showed that averaging improves performance over constant learning rate training (Haegele et al., 2024), though still falling short of LR decay. Nevertheless, its effect on PTQ robustness remains unexplored, despite its simplicity, and compatibility with existing pipelines.

Therefore, we pretrain a 160M-parameter transformer on 100B tokens with a constant learning rate and compare LAtest Weight Averaging (LAWA) (Kaddour, 2022) against several intermediate learning rate cooldowns, with averaging configuration described in Appendix C. As observed in prior work (Ajroldi et al., 2025), in the full-precision setting (Figure 7a), LAWA yields better checkpoints than constant learning rate but does not reach the performance of intermediate cooldowns. In contrast, for 3-bit quantized models (Figure 7b), we find that checkpoints obtained through weight averaging *can match or even surpass* the performance of those trained with learning rate decay.

Finally, we apply the same technique to training trajectories of open-source models. Specifically, we consider OLMo-1B (Groeneveld et al., 2024), averaging checkpoints during training and using LAWA as aggregation scheme (Figure 24). Despite the lack of control over checkpoint saving frequency, the averaged model still improves upon the final one, performing better both in full-precision and after quantization, confirming averaging as a promising direction to improve PTQ robustness.

## 5.3 WEIGHT DECAY

Learning rate and weight decay are coupled in popular AdamW implementations (Paszke et al., 2019). We analyze the impact of changing the weight decay $\lambda$ on the quantization error for a fixed training recipe, with an implementation where learning rate and weight decay $\lambda$ are decoupled (Schaipp, 2024). In Figures 19b and 19c we observe that among models that achieve a comparable performance (seen in the x-axis) in full-precision quantized validation loss, those with larger weight decay $\lambda$ exhibit lower 4- and 3-bit quantization error. This shows that, for $\lambda$ configurations that achieve comparable loss, higher values are preferable to reduce PTQ errors, which confirms Ahmadian et al. (2023) observations. Moreover, compared to Figure 6 we see that changes in $\lambda$ have smaller effect on quantization error than learning rate.

## 6 GEOMETRIC PROPERTIES OF THE LOSS

The findings presented in Section 5 reveal several important relationships between interventions and downstream performance, but is there an underlying, unifying mechanism? To investigate, we

---

[1]We distinguish between *model soups* (Wortsman et al., 2022), which average models from different training runs, and *weight averaging* (Izmailov et al., 2018), which aggregates checkpoints along a single trajectory.

analyze the geometric properties of the loss landscape to illustrate the interaction between these seemingly disconnected phenomena.

## 6.1 LOSS LANDSCAPE

We visualize a 2D slice of the loss landscape (Goodfellow et al., 2015; Li et al., 2018) defined by three checkpoints of interest, $\Theta_K$ the model at the end of training, $\Theta_{K-1}$ the model at a previous step of training, and [2] $\hat{\Theta}_K$, the model at the end of training quantized. We refer to Section F for additional details.

Our goal is to analyze how hyperparameter decisions during pretraining result in different local neighborhoods *around* $\Theta_K$ and $\hat{\Theta}_K$ in the landscape of the loss via the 2D slice they span. In Figure 8 we present four different landscapes, corresponding to pretraining our usual 160M parameter model with different learning rates, as shown in Figure 6. In Figure 8, $\hat{\Theta}_K$ is the result of 4-bit GPTQ quantization, we refer to Figure 25 for analogous results on 3-bit GPTQ quantization. We begin by observing that, as expected, the smaller the learning rate, the closer $\Theta_{K-1}$ and $\Theta_K$ are. Perhaps more interestingly, the distance between $\Theta_K$ and $\hat{\Theta}_K$ follows the same trend, it is larger for larger learning rates. All the slices depict a local minimum around $\Theta_K$.

What is interesting is that we see that in all examples, the landscape is structured similarly in the y-axis, the quantization direction, to the x-axis, the direction to the previous optimization step. In this sense, the geometry of the quantized model seems closely related to the geometry induced by training. Furthermore, the learning rate magnitude is proportional to the flatness of the basin of the loss, where, even though $\Theta_K$ and $\hat{\Theta}_K$ are closer for smaller learning rates, the sharpness of the basin is such that $\hat{\Theta}_K$ falls in a higher loss level, a phenomenon which is exacerbated further for larger weight perturbations e.g. for even lower bit quantization Figure 25.

## 6.2 CURVATURE

To better understand the topology of the loss landscape and the dramatic effect of learning rate decay on quantization robustness, we further examine the second order information of the loss. We estimate the *trace* of the Hessian via Hutchinson estimator (Hutchinson, 1989), and the *sharpness* (maximum eigenvalue) via power iterations, using `PyHessian` (Yao et al., 2019). We refer to Appendix G for details on the estimation procedure and additional results.

In Figure 9 we report the sharpness and trace evolution during the stable and decay phases when training a 160M transformer on 100BT. The maximum eigenvalue shows a consistent rapid surge whenever the learning rate decays. Although we also observe an initial increase in sharpness under a constant step size, a more detailed analysis shows a clear distinction between the two regimes: in the stable phase, only the top eigenvalue initially rises while the others remain small, whereas in the decay phase all eigenvalues increase, underscoring a notable difference between these training dynamics. The trace presents a similar pattern, remaining stable under a constant learning rate, and rising abruptly as it decays, remarkably mirroring the evolution of quantization error in Figure 4.

---

[2]We visualize checkpoints that are trained for 100 billion tokens during $K = 190000$ steps. We save the checkpoints every 2000 tokens, therefore $K - 1 = 188000$.

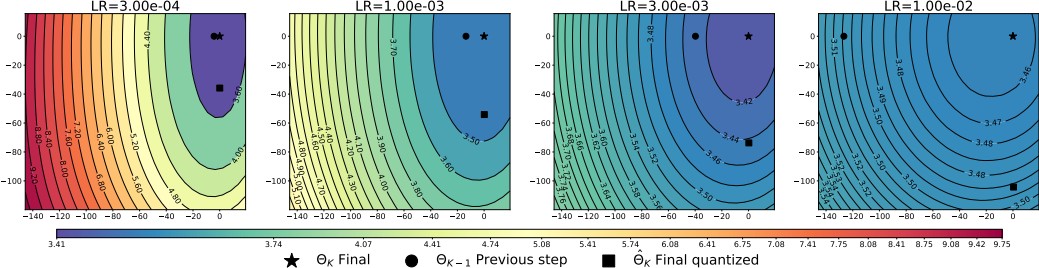

**Figure 8: Landscape of the loss**. We visualize the landscape of the loss in the plane spanned by the weights $\{\Theta_K, \Theta_{K-1}, \hat{\Theta}_K\}$ for learning rates corresponding to the experiment in Figure 6. We observe that flatness of the loss basin is proportional to learning rate magnitude.

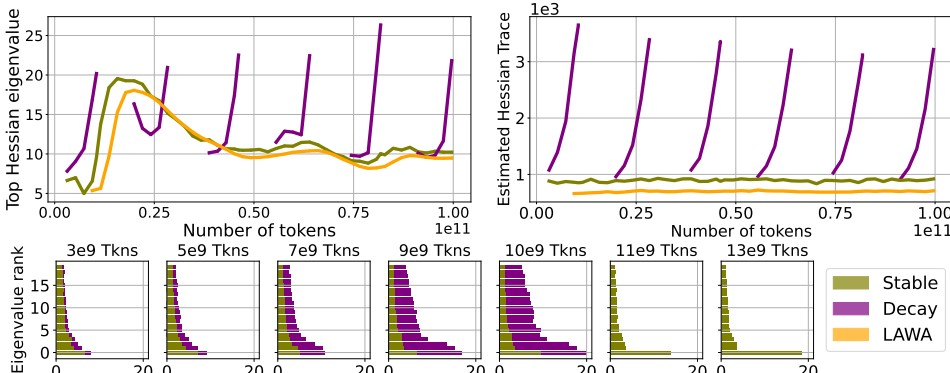

**Figure 9: Sharpness (top left), Hessian trace (top right) and first 25 eigenvalues (bottom)** estimated on the training trajectory of a 160M transformer model (training runs in Figure 4). Sharpness consistently increases when the learning rate decays. Under a constant learning rate, only the top eigenvalue briefly increases while the rest of the spectrum remains low; the second row shows the distribution during this early increase. The trace shows a clearer trend, although it is confounded by being the sum of all eigenvalues.

Although learning rate dynamics are known to affect the Hessian spectrum in simpler settings (Cohen et al., 2025), there is limited understanding of any causal structure in more complex training setups. Based on the observed phenomena, we hypothesize that, as the learning rate decays, the model traverses a sharper region of the loss landscape, *making it more sensitive to perturbations such as quantization.*

Our analysis also indicates that averaging weights during training leads to wider minima, in line with Izmailov et al. (2019). Such improved conditioning of the Hessian might explain the superior quantization robustness of LAWA in Figure 7, but also offers a new perspective on weight averaging: whereas prior work linked it theoretically and empirically to learning rate decay (Sandler et al., 2023), we show that the two methods produce solutions with substantially different curvature properties. We believe that the improved quantization robustness of model soups in Figure 2 may be explained by similar curvature properties induced by souping.

Finally, the benefit of larger learning rates on stochastic gradient descent is well documented (Barrett & Dherin, 2020; Lewkowycz et al., 2020; Gilmer et al., 2022), and it has been suggested that the additional noise leads to *flatter minima*, which should generalize better (Hochreiter & Schmidhuber, 1997; Chaudhari et al., 2017), and require fewer bits to be specified (Hochreiter & Schmidhuber, 1994). When considering training trajectories under different maximum LR (Figure 6), we indeed find that larger ones produce lower sharpness (Figure 26a) and smaller trace estimates (Figure 26b), suggesting the presence of flatter minima, yet interestingly also leading to lower quantization error.

## 7 DISCUSSION

We conduct a systematic investigation of how training interventions affect quantization degradation in language models under controlled experimental configurations. First, we observe that the magnitude of the learning rate determines quantization robustness when all other hyperparameters remain fixed. Second, we identify that averaging checkpoints, either across different data configurations via model souping or along the training trajectory, promotes robustness to quantization. These concrete examples, where quantization degradation noticeably shifts with training dynamics, lead us to advocate studying quantization robustness during routine hyperparameter tuning. We then study geometric properties of the loss to investigate how learning rate and weight averaging affect quantization performance, finding that these interventions coincide with convergence to flatter minima, which we argue might benefit quantization robustness.

Overall, we end on an optimistic note. Our findings indicate that quantization degradation stems from an intricate relationship between training dynamics alluding to general model robustness. As a result, we find that, rather than being an unavoidable consequence of training data scale, it can be acted upon with existing tools, which are especially beneficial for low-bit quantization.

## ACKNOWLEDGMENTS

JG acknowledges the support of the Hector foundation. JG and ACT acknowledge the support of the Amazon Science Hub Tübingen. This research was partially supported by the European Commission under the grant No. 101195233 (OpenEuroLLM).

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

## A QUANTIZATION PROTOCOL

**Alternative quantization methods.** Our results are centered around GPTQ Frantar et al. (2023) a popular and accessible quantization method that works off-the-shelf for new models with minimal engineering overhead. To assess whether the phenomena we observe are specific to GPTQ or reflect broader trends in PTQ, we replicate Figure 4 with LLM.int8() Dettmers et al. (2022) and AWQ Lin et al. (2024). As shown in Figure 10, we observe a consistent association between learning rate driven training dynamics and quantization error.

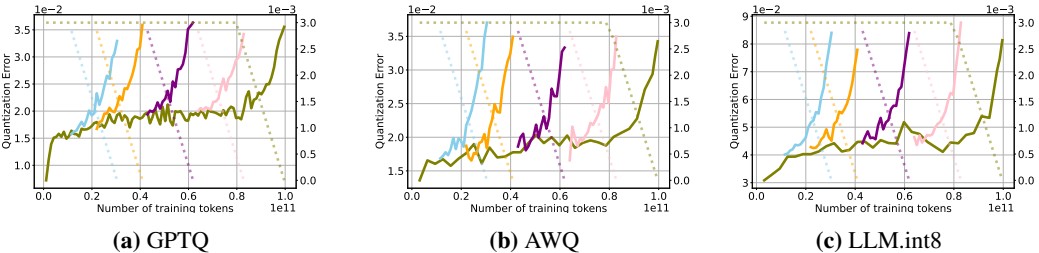

(a) GPTQ      (b) AWQ      (c) LLM.int8

**Figure 10: Quantization error on different 4-bit quantization backends**. We replicate results from Section 4.1, training a 160M-parameter transformer with different quantization backends, and recover similar trends in quantization error during both the constant and cooldown phases of the learning rate schedule.

**Quantization details.** For each model, we quantize the linear layers following the default settings of GPTQModel (ModelCloud.ai & qubitium@modelcloud.ai, 2024) and HuggingFace's internal quantization backend. For GPTQ, we follow common practice (Wolf et al., 2020) and use C4 (Raffel et al., 2023) as the calibration dataset, with a group size of 128. For AWQ (Lin et al., 2024), we use Kwon et al. (2023).Finally, for LLM.int8() Dettmers et al. (2022) we follow HuggingFace Wolf et al. (2020) implementation.

## B PTQ ROBUSTNESS ON ADDITIONAL MODELS IN THE WILD

In this section we report the quantization degradation for additional model families. Although most models follow a regular pattern, some exhibit unpredictable behaviors. Amber (Liu et al., 2023) in Figure 12 displays a brief spike in full-precision validation loss, while the full-precision model recovers, 4-bit PTQ degradation rises sharply, hinting at a change in the training dynamics whose cause we cannot identify. Additionally, Apertus (Apertus Team, 2025) in Figure 15 exhibits very large, fluctuating quantization errors from the beginning, which may indicate numerical issues either in the quantization process or in the weights. However, we note that, even for these models, quantization degradation increases as the learning rates decays, consistent with our previous findings.

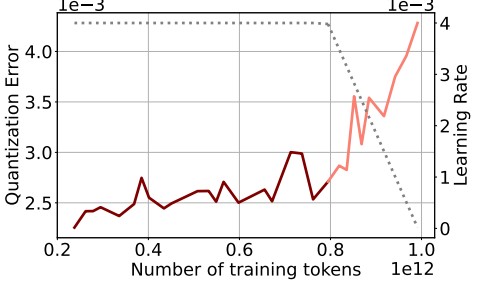 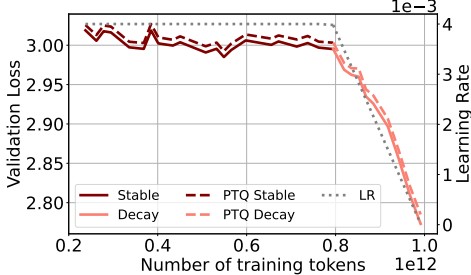

(a) 4-bit PTQ error vs training tokens.      (b) Validation loss vs training tokens.

**Figure 11: Evolution of quantization error and validation loss on OpenSci-1.3B** model (Nezhurina et al., 2025) trained on 1T tokens from Nemotron-cc (Su et al., 2025).

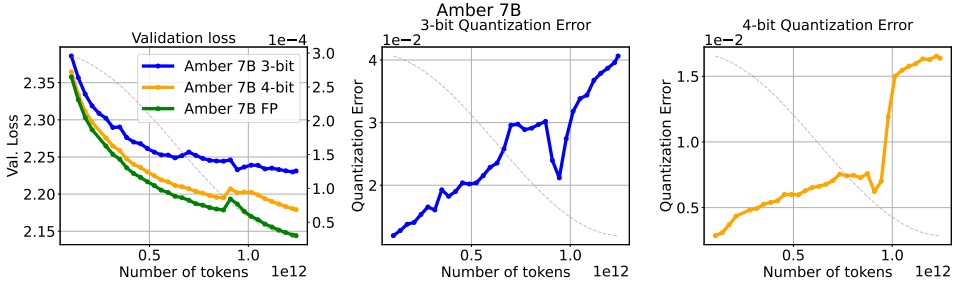

Figure 12: **Quantization degradation for Amber-7B**. 3 and 4-bit quantization with GPTQ.

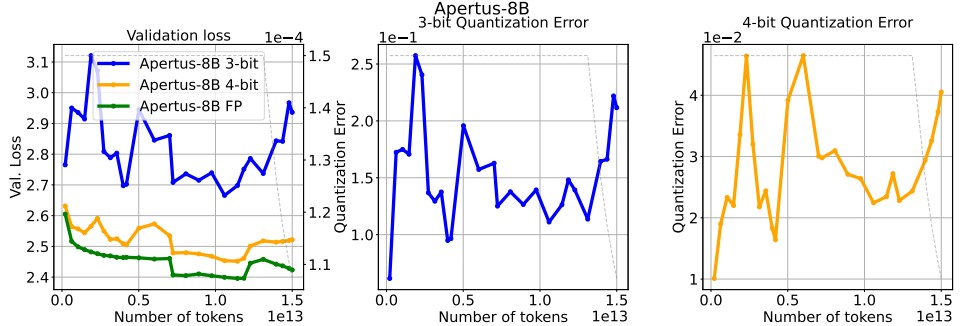

Figure 13: **Quantization degradation for Apertus-8B**. 3 and 4-bit quantization with GPTQ.

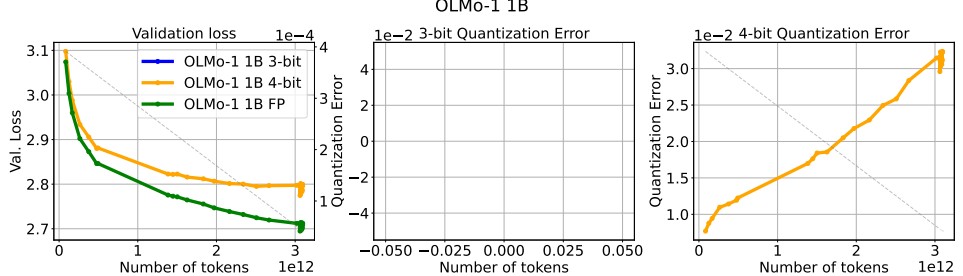

Figure 14: **Quantization degradation for OLMo-1 1B**. 3 and 4-bit quantization with GPTQ.

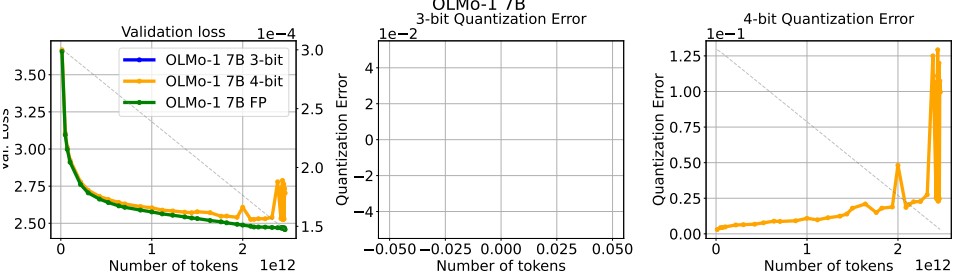

Figure 15: **Quantization degradation for OLMo-1 7B**. 3 and 4-bit quantization with GPTQ.

## C   PRETRAINING HYPERPARAMETERS AND SETUP

**Hyperparameter details.**   We use the open source codebase from Ajroldi (2024) to pretrain
Pythia-160M parameter transformer models (Vaswani et al., 2023; Biderman et al., 2023) on causal
language modeling, training up to 100 billion tokens of FineWebEdu (Penedo et al., 2024) on up
to 8xA100-80GB GPUs. We employ a sequence length of 2048 and batch size of 0.5M tokens.
We use cross-entropy loss and employ Adam (Kingma & Ba, 2014) with decoupled weight decay
(Loshchilov & Hutter, 2019) of 0.1 and gradient clipping of 1, and $\beta_1 = 0.9$, $\beta_2 = 0.95$. For the ex-
periments in Figure 4 we use a WSD learning rate schedule with peak learning rate of 3e-3, warmup
of 1900 steps (1%), and a cooldown duration of 1900 steps (10% of total duration), decaying the
learning rate to zero (Bergsma et al., 2025).

**Weight Averaging.**   For the analysis in Section 5.2 and Figure 7 we use LAtest Weight Averaging
(Kaddour, 2022), collecting checkpoints every 500 optimization steps, and maintaining a rolling
window of length 5 over which weights are uniformly averaged. For the analysis in Figure 24 where
checkpoints are only available at fixed release intervals, we instead average the consecutive released
checkpoints, reporting results for different window lengths.

## D   EVALUATION

Evaluating model performance is influenced by many factors, and quantization methods add an-
other: the calibration dataset. For example, a model quantized using web data for calibration, may
perform better on web-based tasks. In general, interactions between training data, calibration sets,
and validation sets may create complex effects that affect the reliability of results.

To address this problem, we evaluate using two approaches:

- A held-out split of RefinedWeb (Penedo et al., 2024), to gather validation loss performance.
- Downstream performance on the following tasks:
  - **ARC-Challenge (ARC_C)** (Clark et al., 2018)
  - **ARC-Easy (ARC_E)** (Clark et al., 2018)
  - **OpenbookQA (OBQA)** (Mihaylov et al., 2018)
  - **PIQA** (Bisk et al., 2020)
  - **HellaSwag (HSwag)** (Zellers et al., 2019)
  - **WinoGrande (WinoG)** (Sakaguchi et al., 2019)
  - **MathQA** (Amini et al., 2019)
  - **PubMedQA** (Jin et al., 2019)
  - **SciQ** (Welbl et al., 2017)
  - **Social IQa (SIQA)** (Sap et al., 2019)
  - **CommonsenseQA (CSQA)** (Talmor et al., 2019)
  - **MMLU** (Hendrycks et al., 2021)

We evaluate models using LM-eval-harness (Gao et al., 2021) and vLLM (Kwon et al., 2023).
We report per-task accuracy of SmolLM3 in Figures 16, 17, **??** for the full-precision, 3-bit GPTQ
quantzied and 4-bit GPTQ quantized weights respectively.

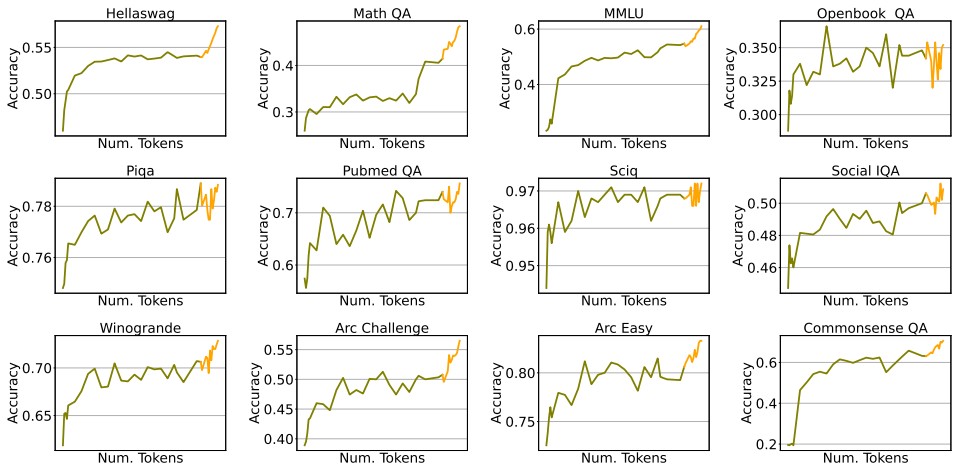

**Figure 16:** SmolLM3 per-task *full-precision accuracy*, measured throughout training.

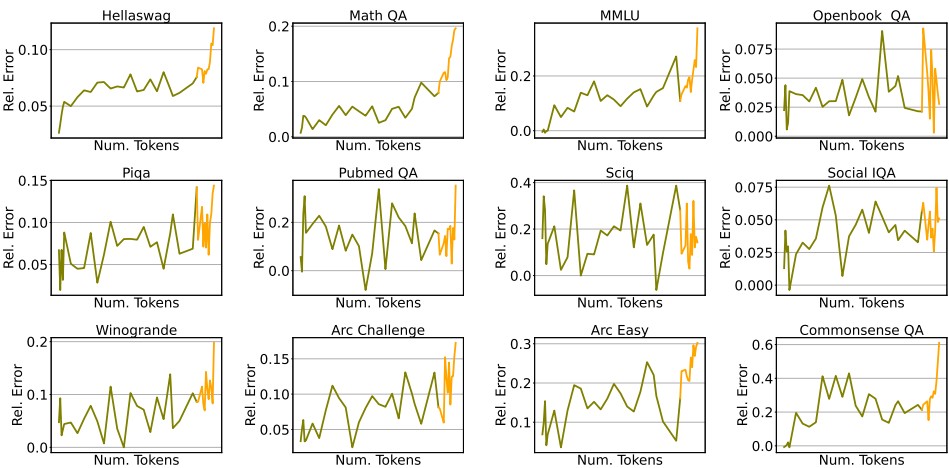

**Figure 17:** SmolLM3 per-task *relative accuracy degradation* under 3-bit GPTQ, measured throughout training.

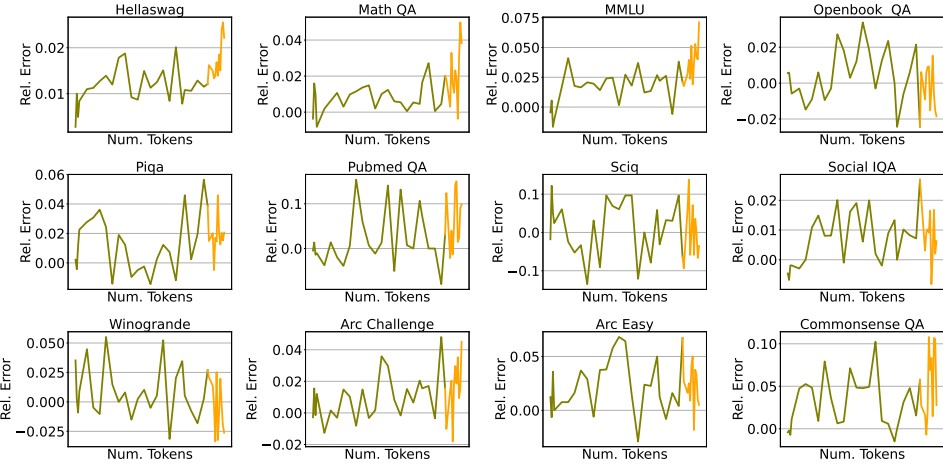

**Figure 18:** SmolLM3 per-task *accuracy degradation* under 4-bit GPTQ, measured throughout training.

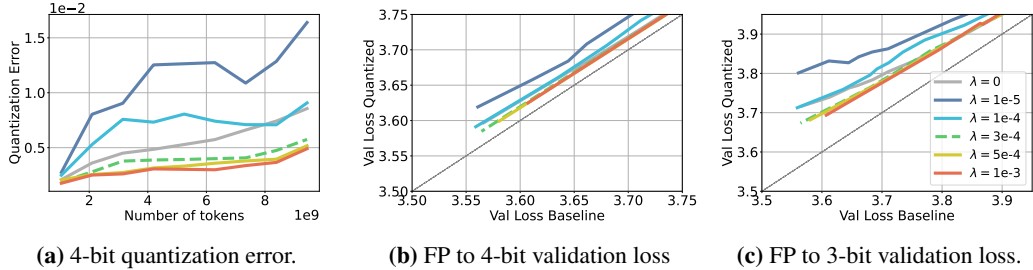

**(a)** 4-bit quantization error.  **(b)** FP to 4-bit validation loss  **(c)** FP to 3-bit validation loss.

**Figure 19: Weight decay promotes PTQ robustness**. With fixed learning rate $3e^{-3}$ and WSD we train several models changing the weight decay parameter $\lambda$ only. We observe that larger $\lambda$ parameters lead to models with higher PTQ robustness. The dashed line represents the $\lambda$ parameter chosen for all prior experiments.

## E    ADDITIONAL RESULTS

In this section we provide additional figures for Section 5.

### E.1    WEIGHT DECAY

We show Figure 19.

### E.2    GRADIENT OF THE LOSS

Recent work has shown that the gradient of the loss increases during the end of training (Defazio, 2025). We have observed that this phenomenon coincides with the decay phase of WSD, to this end, we analyze whether this change in the training dynamics is driving quantization degradation in Figure 20. Fixing all other hyperparameters (more details in Appendix C) we train with AdamW (Loshchilov & Hutter, 2019) (in cyan), and AdamC (Defazio, 2025) (in orange) which aims to correct this behavior. We observe that AdamC reduces the spike of the norm of the loss gradient in Figure 20b while simultaneously changing the norm of the weights in Figure 20c. However, despite modulating different actors of the training dynamics, both optimizers demonstrate almost identical quantization degradation in Figure 20b, suggesting that the norm of the gradient of the loss does not impact quantization performance as a standalone factor, indicating a more complex relationship.

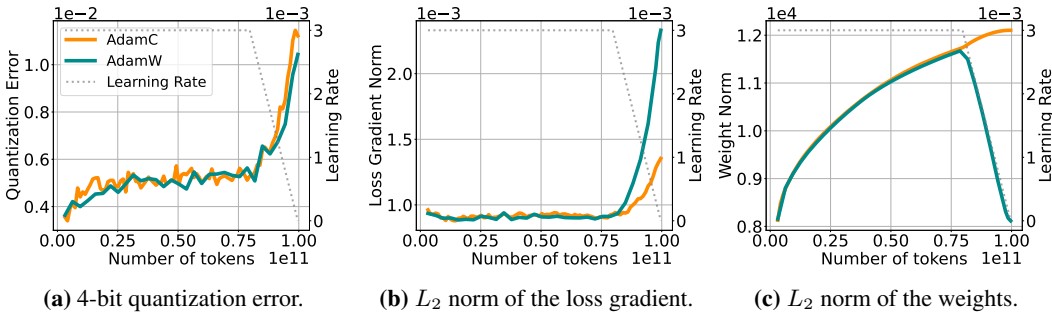

**(a)** 4-bit quantization error.  **(b)** $L_2$ norm of the loss gradient.  **(c)** $L_2$ norm of the weights.

**Figure 20: Loss gradient norm does not directly modulate quantization error**. Quantization error, $L_2$ norm of the loss gradient, and $L_2$ norm of the weights for a 160M model trained with AdamW (Loshchilov & Hutter, 2019) (in cyan) and AdamC (Defazio, 2025) . In Figure 20b we observe that the gradient of the loss spikes during the later iterations when using AdamW, whereas AdamC reduces the spike at the end of training. Furthermore, in Figure 20c we observe that AdamC affects the norm of the weights.

### E.3    COSINE DECAY VS WSD

In Figure 21 we present the quantization error and validation loss for 160M parameter models trained on different token budgets with the same learning rate with cosine decay and with WSD learning rate schedules. We observe that even though quantization error appears to be related to training data

budget for cosine decay learning rate schedule, on WSD quantization error and training data budget appear to be less entangled.

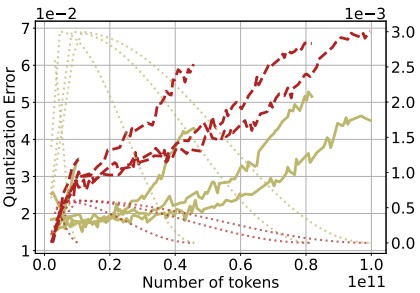

(a) Quantization error vs training tokens.

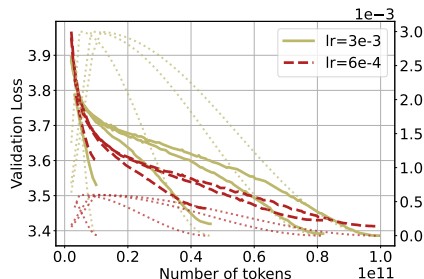

(b) Validation loss vs training tokens.

**Figure 21: PTQ error at different training durations with cosine decay**. We repeat the experiment in 4.1 and Figure 4 with a cosine learning rate schedule. PTQ error (left) varies with training horizon, but peak learning rate and scheduler shape have a larger impact.

### E.4 LEARNING RATE

We repeat the experiment in Section 5.1 on a larger scale, using OLMo2-7B evaluating quantization error during a learning rate annealing run of 50B tokens after the model was pretrained for 250B tokens on 4 different learning rate values. In Figure 23 we observe that, even though the quantization degradation is lower, the same patter arises, where larger learning rates lead to lower quantization degradation, even at the same validation loss.

## F ADDITIONAL DETAILS AND RESULTS FOR LOSS LANDSCAPES

Given a parametric model $\Theta \in \mathbb{R}^{n}$[3], a set $\mathcal{D} := \{(x_i, y_i)\}_{i=1}^{m}$ of feature vectors with corresponding labels pairs, and a loss function $\mathcal{L}(\Theta) = \frac{1}{m} \sum_{i=1}^{m} \ell(x_i, y_i; \Theta)$, we adapt Goodfellow et al. (2015); Li et al. (2018) to visualize a 2D slice of the loss. Our aim is to interpolate the loss between three checkpoints of particular interest, $\Theta_K$ the model at the end of training, $\Theta_{K-1}$ the model at a previous step of training[4], and $\hat{\Theta}_K$, the model at the end of training quantized. Setting $v$ and $u$ as the direction vectors from $\Theta_K$ to $\Theta_{K-1}$ and $\Theta_K$ to $\hat{\Theta}_K$ respectively, and the validation set $\mathcal{D}$, we care about

$$f(\alpha, \beta) = \mathcal{L}(\mathcal{D}; \Theta_K + \alpha v + \beta u) \tag{1}$$

To populate the contour plots we simply sample 1000 points on a regular grid contained by largest bound from the set that we are comparing, and then reconstruct a model from the vectorized definition that we sampled.

To vectorize a quantizaed model, we first "dequantize" by explicitly multiplying the scales and low-bit primitives, and we retrieve a high-precision approximation of the quantized model that we can use.

**3-bit GPTQ Loss Landscape** Analogous to Figure 8, we show the loss landscape for 3-bit GPTQ quantization on Figure 25. We observe that the same pattern occurs, with larger weight perturbations, where the flatness of the basin of the loss is more relevant.

## G SECOND ORDER STATISTICS

**Trace.** In order to approximate the Hessian trace, we can exploit the following result. Let $A \in \mathbb{R}^{n \times n}$ be a symmetric matrix, let $z$ be a multivariate random variable in $\mathbb{R}^n$ with mean $\mu$

---

[3]We visualize 160M parameter models where $n = 1.6e^8$.

[4]We visualize checkpoints that are trained for 100 billion tokens during $K = 190000$ steps. We save the checkpoints every 2000 tokens, therefore $K - 1 = 188000$.

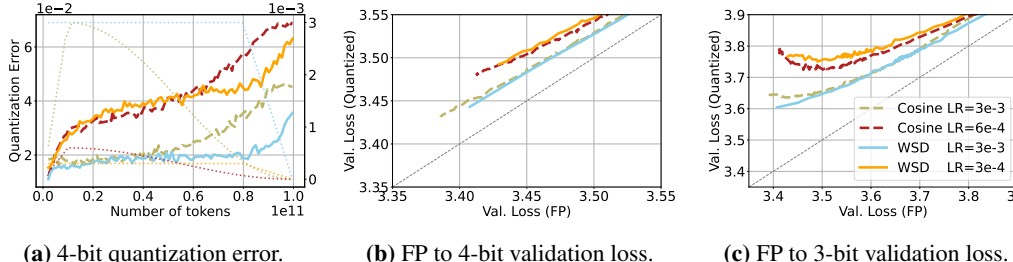

**(a)** 4-bit quantization error.  **(b)** FP to 4-bit validation loss.  **(c)** FP to 3-bit validation loss.

**Figure 22: Warm up-Stable-Decay and Cosine decay**. Figure 22a shows the quantization degradation that results from changing the learning rate magnitude and schedule. We observe that learning rate modulates quantization error regardless of the schedule. Finally, in Figure 22c we observe that cosine schedules have a sharper trade-off in the validation loss of the full precision to the quantized weights.

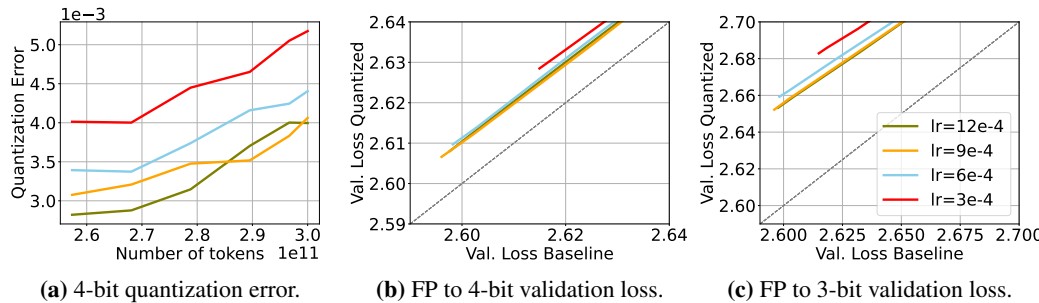

**(a)** 4-bit quantization error.  **(b)** FP to 4-bit validation loss.  **(c)** FP to 3-bit validation loss.

**Figure 23: Larger learning rates lead to lower quantization error**. Figure 23a displays the quantization error achieved by fixing the training recipe and varying the learning rate of OLMo2-7B. We observe that quantization error decreases when employing higher learning rates. Furthermore, Figure 23b and 23c show that, at similar validation loss, larger learning rates achieve better low-bit quantization at no apparent cost.

and covariance $\Sigma$, then:

$$\mathbb{E}[z^T A z] = tr(A\Sigma) + \mu^T \Sigma \mu,$$

where $\mathbb{E}$ indicates the expectation and $tr$ the trace operator. Therefore, for a random vector $z$ with zero-mean and identity covariance matrix, $z^T A z$ is an *unbiased* estimator of $tr(A)$. Hutchinson (1989) showed that when $z$ is distributed accordingly to a multivariate Rademacher distribution, the estimator achieves *lower variance* than choosing $z$ to be a multivariate Gaussian random vector.

We can leverage this property to estimate the Hessian trace of the loss function by drawing samples from a Rademacher distribution and computing Hessian vector products, which can be easily com-

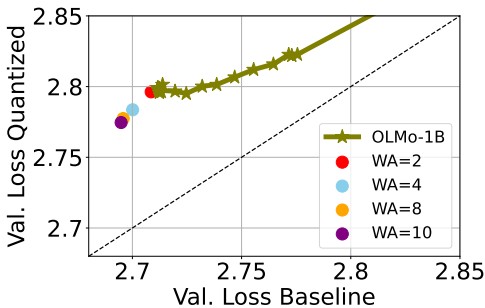

**Figure 24: Weight Averaging improves OLMo performance before and after quantization**. We use LAWA, averaging weights along the OLMo-1B training trajectory. We measure and report validation loss in full precision and after 4-bit quantization. Compared to individual checkpoints on the full trajectory, LAWA yields lower validation loss both before and after quantization, with larger averaging windows performing best.

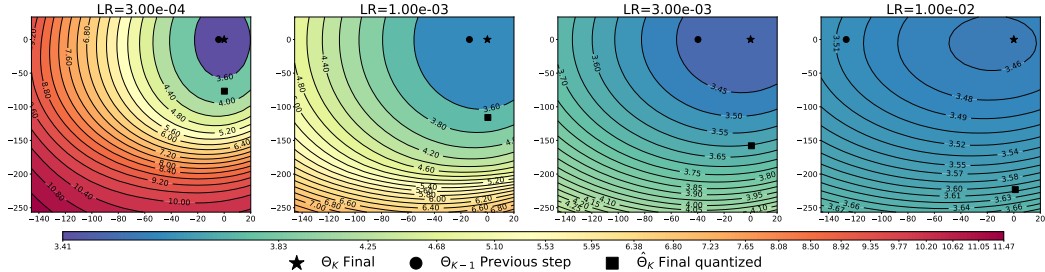

**Figure 25: Landscape of the loss**. We visualize the landscape of the loss in the plane spanned by the weights $\{\Theta_K, \Theta_{K-1}, \hat{\Theta}_K\}$ for learning rates corresponding to the experiment in Figure 6. We observe that flatness of the loss basin is proportional to learning rate magnitude.

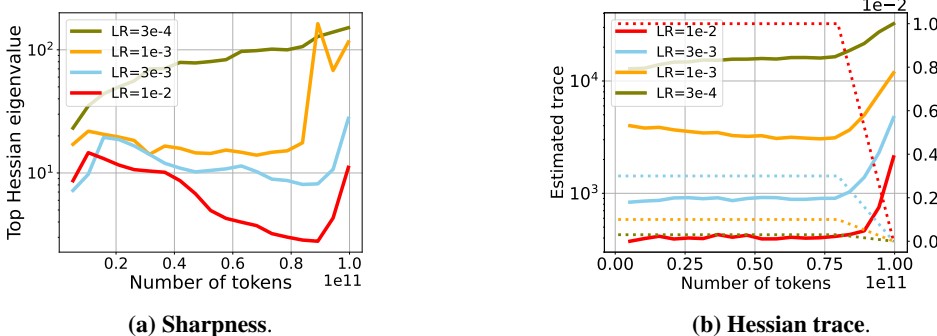

**(a) Sharpness**.  **(b) Hessian trace**.

**Figure 26: Second order statistics across learning rates**. We train using WSD, varying the maximum learning rate, but always decaying it to zero. Higher learning rates lead to lower sharpness and smaller trace estimates, suggesting that the model may have converged to a wider minima. Interestingly, larger learning rate also lead to lower quantization error (Figure 6).

puted with an extra pass over the computational graph. We use `PyHessian` (Yao et al., 2019) for such Monte Carlo estimation in `PyTorch`.

**Sharpness and spectrum.** Furthermore, we measure the largest eigenvalue $\lambda_{max}$ of the Hessian, also referred to as *sharpness*. In order to estimate $\lambda_{max}$ we use power iterations, once again leveraging Hessian vector products computation in `PyHessian`. In some cases we further compute the first 25 hessian eigenvalues.

We measure both summary statistics on in house trained Pythia-160M models. We compute the trace and sharpness of the *validation loss*, computed on an held-out set of 100 text sequences from FineWedEdu, each of length 2048.

## H  LIMITATIONS

Our analysis focuses primarily on the effect of learning rate, schedules, and weight decay leaving other parts of the optimization pipeline unexplored. Factors such as optimizer choice may also affect quantization performance, and we leave the exploration of schedule-free methods (Defazio et al., 2024) to follow-up work. Moreover, although we limit our analysis to dense quadratic model, we expect similar conclusions for sparse (Shazeer et al., 2017) and sub-quadratic architectures (Gu & Dao, 2024).

## DISCLAIMER FOR USE OF LLMS

We primarily used LLMs in coding auto-completion applications to facilitate experimentation. LLMs were also used as writing tools to assist in refining the paper. However, the final version was carefully reviewed and finalized by the authors. No LLMs were used in ideation and experimental design.

