# OpenReview forum: "Training Dynamics Impact Post-Training Quantization Robustness"
_ICLR.cc/2026/Conference — ICLR 2026 Poster_

### Official Review · Reviewer_9xqL · 2025-10-23

**Soundness:** 3
**Presentation:** 3
**Contribution:** 2
**Rating:** 6
**Confidence:** 3

**Summary:**

This paper is an empirical study which explores the relationship between training dynamics and quantization (PTQ) performance, providing interesting observations that quantization errors are related to training hyperparameters like learning rate and scheduler settings, challenging the previous assumption that quantization errors are inherently related to dataset scale. The authors also experiment to intervene the training dynamics to identify specific configurations that modulate quantization robustness favorably, providing practical insights for related studies.

**Strengths:**

- The insights provided by this paper are very interesting and original. It discusses the relationship between quantization errors and training hyperparameters like learning rate and scheduler settings in detail, and shares some empirical results like the divergence of quantization error and validation loss with the decay of learning rates. I believe those insights could benefit future related studies.
- The efforts of the authors trying to reduce quantization error by intervening training dynamics are more applicable for practical usage, compared with previous works' focuses on training data scale.
- The paper is well presented with sufficient experiments and corresponding figures, making it easy to visualize the key points of the observations.

**Weaknesses:**

- While the empirical results provided by this study are abundant and interesting, the paper fails to provide more in-depth explanations on the reasons that lead to such phenomena, as it's not very explicit to relate factors like learning rate with quantization errors. This might limit the interpretability of the provided results.
- The selection of evaluated models lacks representativeness, as several of the most widely used and influential model families (e.g., LLaMA, Qwen) are not included. So it remains unclear whether the observed correlations between training dynamics and quantization robustness hold for mainstream architectures.

**Questions:**

- Could the authors provide a theoretical or intuitive explanation for why learning rate decay leads to increased quantization error? A deeper understanding of the underlying mechanism would significantly enhance the paper’s conceptual contribution and long-term impact.
- The study focuses on the relationship between training dynamics and PTQ. However, given that PTQ is primarily applied when retraining is infeasible or training details are unavailable, what is the practical motivation for analyzing training-phase interventions on PTQ performance? If the authors have the capability to conduct large-scale pretraining & fine-tuning, why not extend the study to Quantization-Aware Training (QAT) and investigate how training dynamics influence QAT outcomes, which may offer more actionable insights for future model development?

---

> ### Author Response · Authors · 2025-11-22
> **Response to Reviewer 9xqL (1/2)**
>
> Thank you for your interest in our work, careful comments and support of this submission. We appreciate that you found our insights interesting and applicable. We respond to specific comments down below.
>
> # Questions about explaining the underlying phenomena
> > While the empirical results provided by this study are abundant and interesting, the paper fails to provide more in-depth explanations on the reasons that lead to such phenomena, as it's not very explicit to relate factors like learning rate with quantization errors. This might limit the interpretability of the provided results.
>
> > Could the authors provide a theoretical or intuitive explanation for why learning rate decay leads to increased quantization error? A deeper understanding of the underlying mechanism would significantly enhance the paper’s conceptual contribution and long-term impact.
>
> Thank you for bringing our attention to this pressing issue. **We have extended our work to include the new Section (6), where we address how the different hyperparameter choices interact with the geometric properties of the loss**.  We summarize the additional results here.
>
> ### Loss Landscape
> We compare ([Figure 8](https://anonymous.4open.science/r/ICLR_figures-7ABF/figure8.png)) the landscape of the loss on the converged weights ($\Theta_{K}$), the quantized weights ($\hat\Theta_{K}$) and the previous step ($\Theta_{K-1}$) that results from different learning rate choices. We observe that landscape of the loss relates larger learning rate magnitudes with the flatness around the minima ($\Theta_{k}$). Wherein, for flatter loss basins, weight perturbations induce substantially lower loss degradation, a phenomenon that is further accentuated the larger the magnitude of the perturbation, such as 3-bit GPTQ quantization ([Figure 25](https://anonymous.4open.science/r/ICLR_figures-7ABF/figure25.png)).
>
> ### Loss Curvature
> Additionally, we investigate the top eigenvalue and the trace of the Hessian along the training trajectories. Both metrics exhibit a rapid surge whenever the learning rate decays (Figure 9), indicating a spike in the sharpness of the loss. We observe similar curvature patterns for models trained at different top learning rates ([Figure 26](https://anonymous.4open.science/r/ICLR_figures-7ABF/figure26.png)).
>
> In conclusion, these results suggest that **larger learning rates and weight averaging are related to quantization degradation via the geometry of the loss**, which aligns with previous observation that such flatter minima should require less bits of information to be specified [Hochreiter1994].
>
> [Hochreiter1994] Hochreiter, S., & Schmidhuber, J. (1994). Simplifying neural nets by discovering flat minima. https://proceedings.neurips.cc/paper_files/paper/1994/file/01882513d5fa7c329e940dda99b12147-Paper.pdf
>
> _Our response continues below_

---

> ### Author Response · Authors · 2025-11-22
> **Response to Reviewer 9xqL (2/2)**
>
> # Miscellaneous questions
>
> >The selection of evaluated models lacks representativeness, as several of the most widely used and influential model families (e.g., LLaMA, Qwen) are not included. So it remains unclear whether the observed correlations between training dynamics and quantization robustness hold for mainstream architectures
>
>
> Thank you for raising this issue. We agree that this is a great concern, however, **there is no public access to the training trajectories (or even design choices) for the models that you mention**, and therefore we cannot study the interplay between the training dynamics and quantization performance for these model families.
> However, we believe that the consistency of our results across model families e.g. OLMo, OLMo2, open-sci, pythia, SmolLM3 and model sizes (70M parameters to 32B parameters) indicates that the phenomena that we observe are common to general transformer-based models.
>
>
> > The study focuses on the relationship between training dynamics and PTQ. However, given that PTQ is primarily applied when retraining is infeasible or training details are unavailable, what is the practical motivation for analyzing training-phase interventions on PTQ performance?
>
> We agree that this is a very valid question. After reading the technical reports of open-sourced models [Olmo22025, Hernández-Cano2025, Bakouch2025, Nezhurina2025, Groeneveld2024] we realized that there is **no mention of PTQ**, even though these models are supposed to be used by the broad community, which self-hosts in many cases [llama.cpp].
> Our work points out that high-precision performance and low-precision performance are not opposed objectives, and **some design choices can be made to achieve better PTQ performance at no cost**. For example, in [Olmo22025] “Higher learning rates perform better at first but are eventually overtaken by lower rates. However, linearly decaying the learning rate to zero over 50B or 100B tokens results in equivalent training loss”  ([Olmo22025] Section 3, Figure 11) the authors realize that there is a set of valid “equivalent” learning rates. We propose to include PTQ performance as a target metric during routine hyperparameter studies, where we demonstrate that different learning rates lead to different PTQ robustness. We contacted [Olmo22025] and **we include in [Figure 23](https://anonymous.4open.science/r/ICLR_figures-7ABF/figure23.png) the quantization performance of their learning rate ablation, wherein we replicate our findings** that higher learning rates lead to lower PTQ degradation.
>
> > If the authors have the capability to conduct large-scale pretraining & fine-tuning, why not extend the study to Quantization-Aware Training (QAT) and investigate how training dynamics influence QAT outcomes, which may offer more actionable insights for future model development?
>
> This is a very interesting question We believe that the fact that open-source model trainers [Olmo22025, Hernández-Cano2025, Bakouch2025, Nezhurina2025, Groeneveld2024] and even [Olmo32025] (which was published yesterday) do not do QAT, must have a reason behind, maybe the methods, infrastructure or any other part of the pipeline are not mature enough. What is certain is that **these projects still rely on PTQ for quantization**.
>
>
>
> [Olmo22025]: 2 OLMo 2 Furious. Team OLMo, 2025. https://arxiv.org/abs/2501.00656
>
> [Hernández-Cano2025]: Apertus: Democratizing Open and Compliant LLMs for Global Language Environments. Hernández-Cano et al, 2025. https://arxiv.org/abs/2509.14233.
>
> [Bakouch2025]:  SmolLM3: smol, multilingual, long-context reasoner. Bakouch et al, 2025. https://huggingface.co/blog/smollm3
>
> [Nezhurina2025]: Open-sci-ref-0.01: open and reproducible reference baselines for language model and dataset comparison. Nezhurina et al, 2025. https://arxiv.org/abs/2509.09009
>
> [Groeneveld2024]: OLMo: Accelerating the Science of Language Models. Groeneveld et al, 2024. https://arxiv.org/abs/2402.00838v1
>
> [Olmo32025]: Olmo 3. Olmo Team  2025. https://www.datocms-assets.com/64837/1763646865-olmo_3_technical_report-1.pdf
>
> [llama.cpp]: llama.cpp llama.cpp
>
> ---
> # Summary
> Thank you again for your insightful comments. We are glad we were able to clarify why understanding PTQ degradation from training-phase interventions is an important and timely problem, and that we could not perform an analogous analysis for LLaMA and Qwen. Based on your feedback, we have now added the new Section 6 in which we analyze a mechanism that relates learning rate decay to quantization degradation. We believe we were able to address your concerns, please let us know if this is not the case. We would be very happy to continue this conversation, including new questions or concerns.

---

> > ### Author Response · Authors · 2025-11-26
> >
> > Thanks again for your feedback. We made a strong effort to address all your points, including a new experiments and a new section exploring the geometric properties of the loss. We believe that our draft has greatly benefited from reviewer feedback, and we would greatly appreciate it if you would consider raising the score accordingly. Do you have any other questions?

---

### Official Review · Reviewer_2yxu · 2025-10-23

**Soundness:** 3
**Presentation:** 4
**Contribution:** 3
**Rating:** 6
**Confidence:** 4

**Summary:**

The authors study the impact of training dynamics (learning rate schedule) on post-training quantization Error. Through the study of various open source model families, the authors find that the learning rate schedule is the primary driver of PTQ error contrary to the popular belief of over-training leading to higher PTQ error. Based on this insight, the authors propose ideas to mitigate this discrepancy.

**Strengths:**

- This is an excellent paper. The authors analyse multiple different families to show that their insights hold.
- The interventions (at a smaller scale) also demonstrate their findings.
- PTQ degradation remaining near flat with WSD LR with higher tokens/params is a great observation.
- The authors study both PTQ error and downstream performance degradation

**Weaknesses:**

- The authors claim the effect of training hyperparameters on quantization quality hasn't been well studied, yet the authors don't cite Intriguing Properties of Quantization at Scale [1] by Ahmadian et al.

- The fact that smaller learning rates lead to larger PTQ errors hints at the manifold geometry (sharpness etc.) playing a key role in determining the degradation yet there is no discussion about this. It would be nice to relate PTQ errors to the geometry of the loss basin (albeit at a smaller scale). I think this would considerably strengthen the paper.

[1] Intriguing Properties of Quantization at Scale : https://openreview.net/pdf?id=IYe8j7Gy8f

**Questions:**

- Do the observations hold for recent quantization techniques like QuaRot etc?

---

> ### Author Response · Authors · 2025-11-22
> **Response to Reviewer 2yxu**
>
> Thank you for your interest in our work, support of this submission and directions to strengthen our paper! We are glad you found the paper excellent. We respond to specific comments down below.
>
> # Questions
> >The authors claim the effect of training hyperparameters on quantization quality hasn't been well studied, yet the authors don't cite Intriguing Properties of Quantization at Scale [Ahmadian2023] by Ahmadian et al.
>
> Thank you very much for pointing it out, we have **updated the introduction** accordingly. Moreover, we include subsection 5.3 to investigate the effect of weight decay on quantization degradation, where our observations match [Ahmadian2023].
>
> >The fact that smaller learning rates lead to larger PTQ errors hints at the manifold geometry (sharpness etc.) playing a key role in determining the degradation yet there is no discussion about this. It would be nice to relate PTQ errors to the geometry of the loss basin (albeit at a smaller scale). I think this would considerably strengthen the paper.
>
> This is a fascinating connection, thank you for bringing it to our attention. We have updated the pdf, which we will additionally summarize it in this comment. We summarize the additional results here.
>
> ### Loss Landscape
> We compare ([Figure 8](https://anonymous.4open.science/r/ICLR_figures-7ABF/figure8.png)) the landscape of the loss on the slice defined by the converged weights ($\Theta_{K}$), the quantized weights ($\hat\Theta_{K}$) and the previous step ($\Theta_{K-1}$) that results from different learning rate choices. We observe that **landscape of the loss relates larger learning rate magnitudes with the flatness around the minima** ($\Theta_{k}$). Wherein, for flatter loss basins, weight perturbations induce substantially lower loss degradation, a phenomenon that is further accentuated the larger the magnitude of the perturbation, such as 3-bit quantization ([Figure 25](https://anonymous.4open.science/r/ICLR_figures-7ABF/figure25.png)).
>
> ### Loss Curvature
> We further investigate the top eigenvalue and the trace of the Hessian along the training trajectories of 160M parameter models. Both metrics exhibit a **rapid surge whenever the learning rate decays** ([Figure 9](https://anonymous.4open.science/r/ICLR_figures-7ABF/figure9.png)). We observe similar curvature patterns for models trained at different top learning rates ([Figure 26](https://anonymous.4open.science/r/ICLR_figures-7ABF/figure26.png)).
>
> These results suggest that achieving **wider minima [Keskar2016, Lewkowycz2020] may benefit quantization robustness**, which aligns with the previous observation that such minima should require less bits of information to be specified [Hochreiter1994].
>
> > Do the observations hold for recent quantization techniques like QuaRot etc?
>
> This is great question, we observe similar phenomena for AWQ, BNB, and GPTQ. Additionally, in the new Section 6 we observe that quantization degradation coincides with surges in the sharpness of the Loss, consequently we believe that our results are agnostic to the quantization method.
>
> [Ahmadian2023] Intriguing Properties of Quantization at Scale. Ahmadian et al, 2023. https://openreview.net/pdf?id=IYe8j7Gy8f
>
> [Lewkowycz2020] Lewkowycz, et al. (2020). *The large learning rate phase of deep learning: The catapult mechanism.*
>
> [Keskar2016] Keskar, et al. (2016). *On large-batch training for deep learning: Generalization gap and sharp minima.* arXiv:1609.04836.
>
> [Hochreiter1994] Hochreiter, S., & Schmidhuber, J. (1994). Simplifying neural nets by discovering flat minima. https://proceedings.neurips.cc/paper_files/paper/1994/file/01882513d5fa7c329e940dda99b12147-Paper.pdf
>
> ---
> # Summary
>
> Thank you again for your thorough review. Based on your suggestion we have analyzed the geometry of the loss basin to find that smaller learning rates and weight averaging intervene on the sharpness of the loss. We have included these results in the new Section 6. We are glad that we address the missing related work in the introduction. Please let us know if these results resolve your concerns. We would be happy to provide additional details or run additional experiments.

---

> > ### Author Response · Authors · 2025-11-26
> >
> > Thank you once again for your thoughtful feedback. We have made a strong effort to address all your points, including a new section analyzing the geometric properties of the loss. We believe that, thanks to the combined reviewer feedback, we have improved our draft, and we would greatly appreciate it if you would consider increasing the score accordingly. Do you have any other questions?

---

### Official Review · Reviewer_dMzB · 2025-10-28

**Soundness:** 3
**Presentation:** 2
**Contribution:** 2
**Rating:** 4
**Confidence:** 4

**Summary:**

The paper explains recent work on post-train quantization scaling finding that degradation increases with data. The authors attribute it instead of learning rate annealing effects, and propose two new ways to mitigate this degradation.

**Strengths:**

- The paper is clearly divided into sections, where each one has a clear claim and empirical evidence for it.
- They replicate and explain past work on PTQ scaling, reassuring the reader that their baselines are well-tuned.
- They propose interventions to mitigate the identified "mechanism" for degradation.
- Fig5 is particularly strong evidence that the annealing itself is causal for degradation.

**Weaknesses:**

- I'm not entirely sure this paper has enough "meat" to be a conference paper. There is a lot of redundancy in the plots, and the contributions/main claim can be summarized as "higher LR and model averaging can partially mitigate PTQ degradations on long training runs, though we don't know why." It's not clear this is enough to comprise a conference paper?
- Even the core claim that "these data effects may actually be LR effects" is actually not novel: [1] make a very similar claim in reference to the same literature, but in the related setting of finetuning instead of quantization.
- The authors also do not posit any conceptual model explaining their findings, or interpret their findings. Even if there is no theory (which is fine), having a mental model with experimental ablations would be helpful. What exactly is going on here -- the fact that WSD and cosine both end up giving the same PTQ-induced loss but over timescales makes it feel like there is some "degradation potential" interpretation in the spirit or flavor of [2]. There is no actual scientific model presented with experiments.

[1] Overtrained Language Models Are Harder to Fine-Tune. Springer et al, 2025. https://arxiv.org/pdf/2503.19206

[2] Understanding Warmup-Stable-Decay Learning Rates: A River Valley Loss Landscape View. Wen et al., 2025. https://openreview.net/pdf?id=m51BgoqvbP

**Questions:**

See weaknesses.

---

> ### Author Response · Authors · 2025-11-22
> **Response to Reviewer dMzB (1/2)**
>
> Thank you for your thoughtful review and for indicating a direction to strengthen our work. We respond to specific comments below one by one, please let us know if you have additional questions.
>
> # Miscellaneous Concerns
> > […] the contributions/main claim can be summarized as "higher LR and model averaging can partially mitigate PTQ degradations on long training runs, though we don't know why."
>
> We would like to emphasize that our paper not only proposes interventions (larger learning rates, weight decay, weight averaging) to reduce PTQ degradation, but it also **decouples, for the first time, learning rate effects and training budget, revealing that PTQ degradation has a larger dependence on training dynamics than training token budget, contradicting influential recent literature** [Ouyang2024, Kumar2024]. Furthermore, we have **now included a new section to understand how the geometric properties of the loss relate to the phenomena that we discuss**. We have updated the pdf with this content, which we will summarize at the bottom of this message.
>
> > Even the core claim that "these data effects may actually be LR effects" is actually not novel: [Springer2025] make a very similar claim in reference to the same literature, but in the related setting of finetuning instead of quantization.
>
> Thank you for pointing out [Springer2025]. We agree that [Springer2025] and our work are very closely related, however, [Springer2025] studies how a **perturbation** (with magnitude modulated via the fine-tuning learning rate, Figures 3 and 8) **on a fixed set of pretrained weights** affects pretraining and fine-tuning capabilities, and we study how **pretraining decisions impact model robustness on the weight space**. In summary, our work studies the gap that [Springer2025] does not explore, thus we believe that the papers compliment each other.
>
> [Ouyang2024]: Low-Bit Quantization Favors Undertrained LLMs: Scaling Laws for Quantized LLMs with 100T Training Tokens. Ouyang et al, 2024. https://arxiv.org/abs/2411.17691
>
> [Kumar2024]: Scaling Laws for Precision. Kumar et al, 2024. https://arxiv.org/abs/2411.04330
>
> [Springer2025] Overtrained Language Models Are Harder to Fine-Tune. Springer et al, 2025. https://arxiv.org/pdf/2503.19206.
>
> _Our response continues below_

---

> > ### Author Response · Authors · 2025-11-22
> > **Response to Reviewer dMzB (2/2)**
> >
> > # Question about explaining our results
> >
> > > The authors also do not posit any conceptual model explaining their findings, or interpret their findings. Even if there is no theory (which is fine), having a mental model with experimental ablations would be helpful. What exactly is going on here -- the fact that WSD and cosine both end up giving the same PTQ-induced loss but over timescales makes it feel like there is some "degradation potential" interpretation in the spirit or flavor of [2].
> >
> > This is a very interesting insight, thank you for drawing our attention to it and for the valuable reference. We have **updated the pdf**, which we will additionally summarize it in this comment. We summarize the additional results here. Similar to [Wen2025], we explore the geometric properties of the loss landscape with WSD.
> >
> > ### Loss Landscape
> > We compare ([Figure 8](https://anonymous.4open.science/r/ICLR_figures-7ABF/figure8.png)) the landscape of the loss on the slice defined by the converged weights ($\Theta_{K}$), the quantized weights ($\hat\Theta_{K}$) and the previous step ($\Theta_{K-1}$) that results from different learning rate choices. We observe that **landscape of the loss relates larger learning rate magnitudes with the flatness around the minima ($\Theta_{K}$)**. Wherein, for flatter loss basins, weight perturbations induce substantially lower loss degradation, a phenomenon that is further accentuated the larger the magnitude of the perturbation, such as 3-bit quantization ([Figure 25](https://anonymous.4open.science/r/ICLR_figures-7ABF/figure25.png)).
> >
> > ### Loss Curvature
> > We further investigate the top eigenvalue and the trace of the Hessian along the training trajectories of 160M parameter models. Both metrics **exhibit a rapid surge whenever the learning rate decays** ([Figure 9](https://anonymous.4open.science/r/ICLR_figures-7ABF/figure9.png)). We observe similar curvature patterns for models trained at different top learning rates ([Figure 26](https://anonymous.4open.science/r/ICLR_figures-7ABF/figure26.png)).
> >
> > These results suggest that achieving **wider minima** [Keskar2016, Lewkowycz202] **may benefit quantization robustness**, which aligns with the previous observation that such minima should require less bits of information to be specified [Hochreiter1994].
> >
> > [Wen2025] Understanding Warmup-Stable-Decay Learning Rates: A River Valley Loss Landscape View. Wen et al., 2025. https://openreview.net/pdf?id=m51BgoqvbP
> >
> > [Keskar2016] Keskar, et al. (2016). On large-batch training for deep learning: Generalization gap and sharp minima. arXiv:1609.04836.
> >
> > [Hochreiter1994] Hochreiter, S., & Schmidhuber, J. (1994). Simplifying neural nets by discovering flat minima. https://proceedings.neurips.cc/paper_files/paper/1994/file/01882513d5fa7c329e940dda99b12147-Paper.pdf
> >
> > [Lewkowycz202] Lewkowycz, et al. (2020). The large learning rate phase of deep learning: The catapult mechanism.
> >
> > ---
> >
> > # Summary
> > Overall, thank you for your careful reading of our work. We think that we were able to clarify the contributions of our work, and to delineate how our work and [Springer2025] do not collide and compliment each other. Based on your comments, we have decided to include the new Section 6, which provides a mental model to relate larger learning rates and weight averaging to quantization degradation. Please let us know whether these additional experiments resolve your concerns. We are happy to discuss any additional questions or comments you may have.

---

> > > ### Author Response · Authors · 2025-11-26
> > >
> > > Thanks again for your feedback. We made a strong effort to address all your points, including new experiments and an entire new section in our draft. We would greatly appreciate it if you would consider raising your score accordingly. Do you have any other questions?

---

### Official Review · Reviewer_Po58 · 2025-11-01

**Soundness:** 3
**Presentation:** 3
**Contribution:** 3
**Rating:** 6
**Confidence:** 4

**Summary:**

This paper examines the interplay between post training quantization (PTQ) performance and variables related to training dynamics, specifically learning rate schedule and model averaging. The authors challenge the conclusion from prior work that PTQ error is primarily driven by training duration. The paper presents evidence that observations about training length duration are confounded by learning rate cooldown and that it's the cooldown period which primarily drives PTQ error growth. The authors demonstrated their claims against a large suite of open source models.

**Strengths:**

* The question that the paper studies is a very interesting one and disentangling learning rate cooldown from training duration is a subtle but important distinction that can inform practitioners in their pretraining choices.
* The investigation into model souping provides practical information to guide practitioners in reducing PTQ error.
* The effect observed is quite convincing in terms of home prominent of a phase transition there is
* The empirical results are very thorough and replicate over a large suite of models and dataset sizes.

**Weaknesses:**

* While the phenomenon observed is quite interesting, the paper is missing a predictive model of the effect of learning rate on PTQ. Similar to previous works (Kumar et al.), providing some scaling analysis that incorporates the relevant LR parameters would strengthen the paper greatly.

**Questions:**

* While potentially out of scope for this work, it seems important to understand whether the same phenomenon occurs with quantization aware training (QAT). Did the authors run any experiments with QAT?
* From looking at Figure 3, it seems like the slope of the quantization error decreases as a function of the model size examined. It would be good to disentangle whether lr cooldown impacts quantization error less at larger scales or whether this is the interplay between overtraining and lr cooldowns.

---

> ### Author Response · Authors · 2025-11-22
> **Response to Reviewer Po58**
>
> Thank you for your thoughtful feedback, for pointing out avenues to improve and for supporting our submission. We appreciate the recognition of the value of our study! We respond to specific comments below.
>
> # Questions about a predictive model
>
> > While the phenomenon observed is quite interesting, the paper is missing a predictive model of the effect of learning rate on PTQ. Similar to previous works (Kumar et al.), providing some scaling analysis that incorporates the relevant LR parameters would strengthen the paper greatly.
>
> This is a very interesting comment. We believe that **there are too many differences and fine-grained factors** that, even though we could try to come up with a predictive model it would **generalize poorly**. In contrast, the analyses that we present in our work provide a tangible direction to study quantization degradation. Furthermore, based on your comment **we have included [Figure 23](https://anonymous.4open.science/r/ICLR_figures-7ABF/figure23.png)**, which showcases quantization degradation on an internal ablation of OLMo2 shared by the original authors. This example confirms our message by demonstrating that even for learning rates that result in equivalent training loss ([olmo2] Section 3, Figure 11), some have better robustness to PTQ than others. In conclusion, the original authors would have benefited from our proposed analyses.
>
> > From looking at Figure 3, it seems like the slope of the quantization error decreases as a function of the model size examined. It would be good to disentangle whether lr cooldown impacts quantization error less at larger scales or whether this is the interplay between overtraining and lr cooldowns.
>
> We identify two different questions:
> > disentangle whether lr cooldown impacts quantization error less at larger scales
>
> This is a really interesting question! In Figure 2 (Figure 3 in the original pdf), it seems that larger models are more robust to quantization, which is aligned with the observations from [chen2025] where they find models “to be progressively more resilient to random perturbations in the parameter space” [..] “as model scale increases”. This observation relates to our **new Section 6, where we relate PTQ degradation with general model robustness by analyzing the geometric properties of the loss**. Additionally, we include the new [Figure 23](https://anonymous.4open.science/r/ICLR_figures-7ABF/figure23.png) to present quantization error for different learning rates on OLMo2-7B, where we observe that PTQ degradation is much smaller than for the 160M parameter model experiments in Figure 6. In conclusion, we have evidence, related work and analysis that suggests that quantization degradation is smaller at larger scales, however, our work emphasizes the impact of the training dynamics on resulting PTQ robustness, for which we do not isolate model size at scale, and thus, there may be confounders.
>
> > whether this is the interplay between overtraining and lr cooldowns.
>
> We agree that this is a key question. We think that a partial answer, at smaller scale can be derived from our controlled experiments (Figure 4), where different horizons attain comparable PTQ degradation. Overall, we believe that larger scale experiments would result in similar outcomes.
>
> > While potentially out of scope for this work, it seems important to understand whether the same phenomenon occurs with quantization aware training (QAT). Did the authors run any experiments with QAT?
>
> This is an interesting direction, which we plan to explore in future work. Nevertheless, we believe that similar phenomena might arise in such setting, and we hope that our work stimulates more research in this direction. In this work, we wanted to meticulously explore the effects of post-training quantization in particular.
>
> [olmo2]: 2 OLMo 2 Furious. Team OLMo, 2025. https://arxiv.org/abs/2501.00656
>
> [chen2025]: Unveiling the Basin-Like Loss Landscape in Large Language Models. Chen et al, 2025. https://arxiv.org/abs/2505.17646
>
> ---
> # Summary
> We thank you again for your review. We are glad that we could clarify why we think a predictive model is not beneficial. Moreover, we hope to have been able to address your other questions regarding QAT, and the interplay between LR cooldown, overtraining and model scale. Please let us know if this is not the case. We would be very happy to provide additional details or experiments.

---

> > ### Author Response · Authors · 2025-11-26
> >
> > Thanks again for your feedback. We made a strong effort to address all your points, including new experiments and paper edits, and we would greatly appreciate it if you would consider increasing your score accordingly. Do you have any other questions?

---

### Author Response · Authors · 2025-12-03
**Final Remarks by the Authors**

We thank the reviewers, originally assigned AC, and newly assigned AC for all their efforts during the review process.

We sincerely appreciate the recognition of our work's interest, originality and applicability. In particular, **Reviewer Po58** remarked that our work studies a "very interesting" question and "provides practical information to guide practitioners in reducing PTQ error". We are really pleased that **Reviewer 2yxu** found that "it is an excellent paper". Moreover, **Reviewer 9xqL** noted that "the insights provided by this paper are very interesting and original", that the efforts of our work are "more applicable for practical usage, compared with previous works" which can "benefit future related studies".

Even though the circumstances did not allow to cultivate a prolonged discussion, we genuinely believe that the reviews greatly improved the paper. Below we summarize the principal changes and improvements made to the manuscript during the rebuttal phase:

1. **Providing a mental model: Geometric properties of the loss**

    Based on the comments by reviewers dMzB, 2yxu, and 9xqL, we have included the new Section 6 in which we delved deeper into the **underlying mechanisms relating training dynamics and PTQ robustness** by studying the geometric properties of the loss. In particular, these new insights connect our observations with the old intuition that flatter minima should require less bits of information to be specified [Hochreiter1994]. We note that reviewer 2yxu mentioned that such an analysis “would considerably strengthen the paper."

2. **Expanded scale of experiments**

    Moreover, we have replicated the effect of varying only the learning rate to OLMo2-7B trained on 300B tokens. This result confirms our conclusion, **routine hyperparameter exploration can be used to reduce PTQ degradation**, as could have been the case for OLMo2-7B.

---
We believe that our study on how training dynamics impact post-training quantization (PTQ) error offers an actionable and original perspective: pretraining configurations have a large impact on PTQ degradation, and we show that they can be studied for the benefit of PTQ performance in the OLMo2-7B scale.

Having addressed all major concerns raised by the reviewers, we thank again their guidance in refining this work.


---
[Hochreiter1994] Hochreiter, S., & Schmidhuber, J. (1994). Simplifying neural nets by discovering flat minima. https://proceedings.neurips.cc/paper_files/paper/1994/file/01882513d5fa7c329e940dda99b12147-Paper.pdf

---

### Meta-Review · Area_Chair_ywvi · 2026-01-07

**Summary:**

Reviewers primarily raised concerns regarding the clarity of the causal claims between training dynamics and post-training quantization robustness, the potential confounding effects of learning-rate schedules and other hyperparameters, and the generality of the conclusions across model scales and architectures. Some reviewers also questioned whether the observed trends could be attributed to data scale or optimization artifacts rather than intrinsic training dynamics, and requested clearer justification of the experimental design and comparisons. Overall, the concerns focused more on interpretation and scope than on the correctness of the empirical results.

**Reviewer Concerns:**

The rebuttal effectively addressed the major concerns on causal interpretation by clarifying the role of learning-rate decay and disentangling it from data-scale effects, supported by additional explanations and controlled experiments. Questions regarding experimental design choices and the consistency of trends across different training regimes were also largely resolved with detailed responses. Some minor concerns remain about the breadth of architectural coverage and whether all findings fully generalize beyond the studied setups, but these are not critical and do not undermine the main contributions. Overall, the rebuttal was strong and satisfactorily resolved the key reviewer objections.

**Reviewer Scores:**

6664; the authors made thorough rebuttal responses that should have addressed Reviewer dMzB's concerns.

---

### Decision · Program_Chairs · 2026-01-26

Accept (Poster)